# Learning with Noisy Labels [Re]visited

## Abstract

Learning with noisy labels (LNL) is a subfield of supervised machine learning investigating scenarios in which the training data contain errors. While most research has focused on synthetic noise, where labels are randomly corrupted, real-world noise from human annotation errors is more complex and less understood. Wei et al. (2022b) introduced CIFAR-N, a dataset with human-labeled noise and claimed that real-world noise is fundamentally more challenging than synthetic noise. This study aims to reproduce their experiments on testing the characteristics of human-annotated label noise, memorization dynamics, and benchmarking of LNL methods. We successfully reproduce some of the claims but identify some quantitative discrepancies. Notably, our attempts to reproduce the reported benchmark reveal inconsistencies in the reported results. To address these issues, we develop a unified framework and propose a refined benchmarking protocol that ensures a fairer evaluation of LNL methods. Our findings confirm that real-world noise differs structurally from synthetic noise and is memorized more rapidly by deep networks. By open-sourcing our implementation, we provide a more reliable foundation for future research in LNL.

## 1 Introduction

Learning with noisy labels (LNL) is a fundamental challenge in supervised machine learning, where errors in training data labels can significantly degrade model performance (Nettleton et al., 2010). Deep neural networks, in particular, are prone to memorizing mislabeled examples (Zhang et al., 2017), leading to overfitting and poor generalization (Xie et al., 2021). Addressing this issue is crucial for real-world applications, where high-quality labeled data is expensive and human annotation errors are inevitable. Most of the research in LNL has focused on synthetic noise, where labels are artificially corrupted under controlled conditions, such as random class swaps. Although this provides a straightforward experimental setup, it fails to capture the complexity of real-world label noise (Jiang et al., 2020).

A wide range of techniques have been developed to mitigate the effects of noisy labels. The problem setup shared among all of them involves training an underlying classifier (e.g., a ResNet (He et al., 2015)) with a learning strategy robust to label noise. Figure 1 illustrates the general setup. Common approaches in the field include using loss functions robust to noise (Ghosh et al., 2017; Zhang & Sabuncu, 2018; Wang et al., 2019), adding specialized regularization (Lukasik et al., 2020; Wei et al., 2022a), adjusting loss based on noise estimations (Patrini et al., 2017; Xia et al., 2019; Reed et al., 2015), filtering noisy samples (Han et al., 2018; Yu et al., 2019; Wei et al., 2020), and adding additional layers to the base classifier (Goldberger & Ben-Reuven, 2017; Bai et al., 2021). For a detailed survey of LNL approaches, we refer the reader to Song et al. (2022).

Despite significant progress, most of these methods have been evaluated under synthetic noise assumptions, which do not fully reflect real-world annotation errors. Unlike synthetic noise, real-world noise tends to be instance-dependent (Chen et al., 2021), correlating with visual similarity between classes and annotator biases (Wei et al., 2022b).

To bridge the gap between synthetic and real-world label noise, Wei et al. (2022b) introduced CIFAR-N, a dataset containing human-annotated label noise based on CIFAR-10 and CIFAR-100 (Krizhevsky et al., 2009). They investigate how real-world noise impacts learning and whether existing LNL methods remain

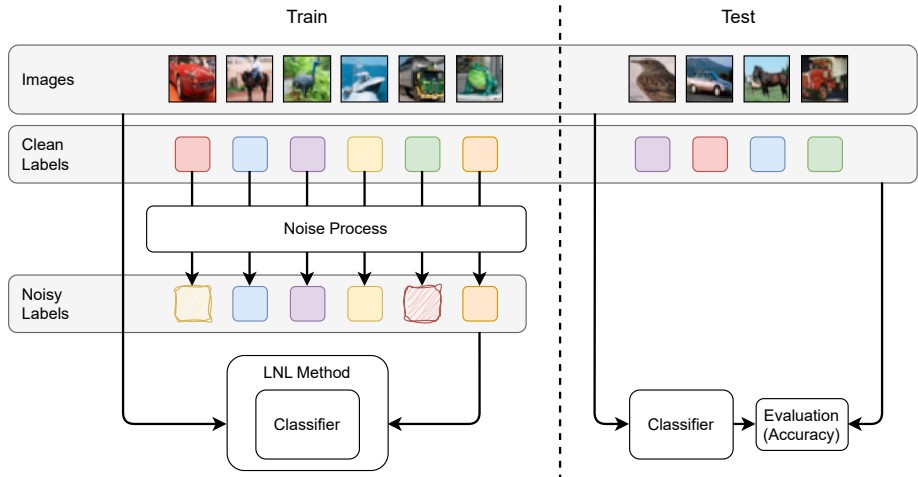

Figure 1: An illustration of a setup for learning with noisy labels. During the training phase (left side), the model is trained using samples with potentially corrupted labels. The LNL algorithm tries to mitigate this noise. During the evaluation phase (right side), the trained model is evaluated with uncorrupted labels.

effective in practical scenarios and claim that learning with real-world noise is fundamentally more challenging than with synthetic noise. Finally, they evaluate 20 state-of-the-art LNL methods on the proposed dataset.

Their key findings suggest that real-world noise differs structurally from synthetic noise, as it is feature-dependent and often reflects human annotation tendencies rather than random corruption. Additionally, deep networks memorize real-world noise more rapidly, overfitting human-labeled noise faster than synthetic noise, which leads to different learning dynamics. Consequently, the benchmark results indicate that real-world noise poses a harder challenge for LNL methods, as models trained on synthetic noise often achieve significantly higher accuracy on clean data than those trained on human-annotated noisy labels.

Despite these contributions, the original work has several limitations. The code used to conduct the experiments is not publicly available, and key methodological details are insufficiently described, making their claims difficult to verify. Moreover, the benchmark results are published on public leaderboards and used as is in subsequent works (Chen et al., 2024; Xiao et al., 2023; Zhang et al., 2025), even though flaws in the testing methodology prevent direct and fair comparisons. To address these issues, we make the following contributions:

1. We reproduce the experiments described by Wei et al. (2022b) and open-source the code, so the experiments can be replicated by others. Our source code is publicly available at (*URL anonymized and submitted as supplementary material*).

2. We implement a framework to benchmark LNL methods under controlled conditions. With it, we are able to establish unified noising and testing pipelines for all methods. To our knowledge, our framework is the first such attempt in the LNL field.

3. We benchmark LNL methods in a controlled environment, fixing the appropriate parameters and enabling a fair comparison of LNL methods on CIFAR-N.

## 2 Scope of Reproducibility

The original paper by Wei et al. (2022b) introduces the CIFAR-N dataset, arguing that real-world human annotation noise differs significantly from synthetic noise and presents a more challenging learning problem. Their findings are based on a series of experiments comparing the characteristics of real and synthetic noisy labels, analyzing the memorization behavior of deep networks, and benchmarking 20, at the time, state-of-the-art LNL methods.

During their analysis of CIFAR-N, the authors make several key observations: (1) the noise distribution in CIFAR-N is imbalanced, favoring similar-looking classes, (2) misannotations primarily occur between visually similar categories, (3) the noise exhibits a mix of symmetric and asymmetric patterns, (4) some noisy labels may reflect multiple valid annotations rather than errors, and (5) learning on real-world label noise is more challenging than learning on synthetic noise. Based on these observations, they make three central claims: (i) **human annotation noise is feature-dependent**, (ii) **deep networks memorize real-world noisy labels faster than synthetic ones**, and (iii) **LNL methods more easily model synthetic noise** than real-world human annotation noise.

In this work, we aim to reproduce and validate these claims by conducting an independent evaluation of the key findings from the original study. Specifically, we investigate:

i. **Differences Between Real-World and Synthetic Noise**: We replicate the hypothesis testing methodology to quantitatively compare human annotation noise with synthetic noise. We assess whether real-world noise is feature-dependent and whether label transitions exhibit different structural properties (Section 3.2).

ii. **Memorization of Noisy Labels by Deep Networks**: We reproduce the memorization effect experiment to examine whether deep neural networks overfit real-world noisy labels faster than synthetic ones. We analyze the impact of different noise levels and label sets on memorization trends (Section 3.3).

iii. **Benchmarking of LNL Methods**: We attempt to reproduce the benchmark results reported for 10 of the 20 LNL methods, evaluating their performance on both real-world and synthetic noise. We investigate inconsistencies in the reported results and assess whether the stated training methodology leads to the same outcomes (Section 3.4).

iv. **Evaluation of Benchmarking Methodology**: We examine whether the benchmarking approach used in the original paper is fair and consistent across methods (Section 3.5).

Going beyond the original study, we propose a revised benchmarking protocol that ensures comparability across different LNL techniques. Throughout our study, we carefully document any deviations from the original methodology and highlight areas where ambiguities in the original paper may have impacted reproducibility. By open-sourcing our implementation (*URL anonymized and work submitted as supplementary material*), we aim to provide a transparent framework for future research in learning with noisy labels.

## 3 Methodology

We follow the authors' described methodology for all experiments except when reproducing their LNL benchmark, which we justify in Section 3.4. Where the procedures are originally not well described or are ambiguous, we describe our selected methodology in greater detail. We also present why their benchmark results should not be used as a benchmark and provide improvements that enable a fairer comparison.

### 3.1 Datasets

We use the CIFAR-N datasets provided by the benchmark authors Wei et al. (2022b). Their work is based on the standard CIFAR-10 and CIFAR-100 datasets (Krizhevsky et al., 2009), consisting of $50,000$ train and $10,000$ test images with image-level class annotations. As the names suggest, CIFAR-10 includes 10 label categories, while CIFAR-100 has 100.

The authors propose noisy extensions for both datasets, which keep the original images but replace the labels with real-world noisy annotations. Only the training sets of both datasets have these, while the $10,000$ testing labels are kept from the original CIFAR-10 and -100 datasets and are considered noise-free. The $50,000$ training labels from the original CIFAR datasets are considered ground truth clean labels.

The CIFAR-10N extension includes five sets of noisy labels obtained from different annotators with varying degrees of noise. Specifically, the authors present the following label sets:

- **Random1, Random2, Random3**: These directly include the labels submitted by different annotators. These have a medium noise level, with 17.23%, 18.12%, and 17.64% noise ratios, respectively.

- **Aggregate**: Labels in this set are obtained from Random1, Random2 and Random3 via majority voting. Whenever there is a 3-way disagreement, the label is selected randomly. Due to the aggregation, the noise level here is the lowest at 9.03%.

- **Worst**: Similarly, as in the *Aggregate* set, labels here are also obtained from the three *Random* sets but are aggregated so that incorrect labels are taken wherever possible. As such, the noise level here is the highest, with 40.21% incorrect labels.

Clean-to-noisy label transition matrices in Figure 2 indicate percentages of misassigned labels and common mistakes (e.g., mislabeling truck as an automobile).

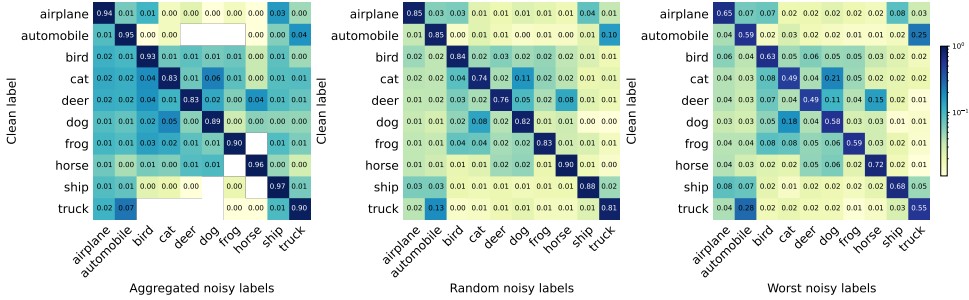

Figure 2: **Transition matrices** of CIFAR10-N for Aggregate, Random1 and Worst label sets. A single row denotes the percentages of transitions of the row label to any of the column labels. Missing numbers indicate that no such transition was found in the provided noisy label set. The color bar is log-norm transformed to highlight noise levels.

Similarly, the CIFAR-100N extension includes two sets of real-world noisy labels. Besides the original 100 categories, the authors also present a new set of coarse annotations of 20 super-classes to help the annotators during annotation. The annotators are first asked to assign a super-class, from which they then later select the fine-grained class. The two noisy label sets are then defined as follows:

- **Coarse**: consists of the 20 super-class labels with a noise rate of 25.60%.

- **Fine**: consists of all 100 fine categories and has 40.20% of noisy labels.

The authors observe that in CIFAR-100N some label noise may not necessarily indicate incorrect annotations but rather the presence of multiple valid labels within an image. We visualize some examples in Figure 3.

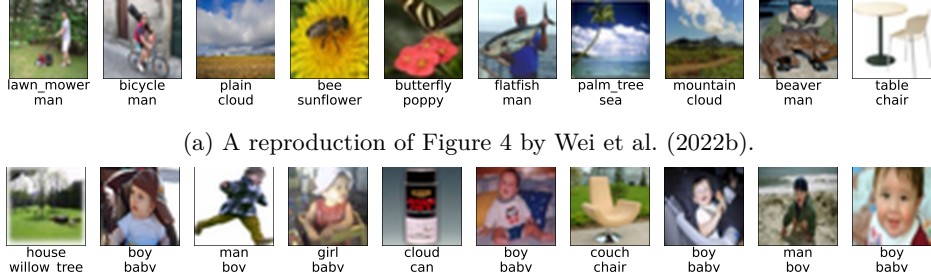

(a) A reproduction of Figure 4 by Wei et al. (2022b).

(b) Images to which their noisy labels might be better suited than their original labels.

Figure 3: **Examples of training images with multiple valid labels** from CIFAR-100N. The first caption contains the clean CIFAR-100 label, and the bottom one is the human-annotated "noisy" label.

They provide examples of such cases, with the frequent case being a fisherman holding a flatfish in his hands. We reproduce their exemplar images in Figure 3a. During our analysis, we even observed that some noisy annotations actually describe the images better than their original labels.

## 3.2 Reproducing Noise Hypothesis Testing

Wei et al. (2022b) qualitatively and quantitatively show that real-world human noise differs from synthetic noise with hypothesis testing. We follow the same procedure while taking note of certain ambiguous decisions.

We start by fitting a ResNet-34 model on clean CIFAR-10 train labels. The authors do not explicitly explain how the model was trained, so we use the default training methodology presented in Section 3.4. Once trained, we use the classifier to get image embeddings for all $50,000$ training images.

Next, for each of the $N = 10$ classes, we cluster its embeddings into $\nu = 5$ clusters $C_{n,v}, \forall n \in N, \forall v \in \nu$ using a K-Means model. As all cluster members belong to the same clean class, we can then obtain a $N$-dimensional transition vector for each cluster: $\mathbf{p}_{n,v} = [\mathbb{P}(\widetilde{Y} = j | Y = n, X \in C_{n,v})]_{j \in [N]}$. We do this for both real-world and synthetic labels, specifically the Random1 label set and its synthetic counterpart. We generate the synthetic label set by sampling from an asymmetric transition matrix estimated from the CIFAR-N noisy labels. Following the approach in the original paper, we compute this matrix by counting class-wise label flips between the human-annotated (noisy) labels and the clean labels. The resulting transition matrix (visualized in Figure 2) is then used to synthesize new noisy labels, producing a synthetic counterpart following the same transition matrix but without instance-dependence. At this point, we plot the resulting transition vectors in each class and cluster and qualitatively check for differences between the real-world and synthetic transition vectors.

We repeat the previous steps $E = 10$ times, generating a new set of synthetic labels and embeddings each time. To get different embeddings, we randomly augment the images with horizontal flips and random crops. This gives us a set of transition vectors $\{\mathbf{p}_{e,k,v}\}_{e \in [E], k \in [K], v \in [\nu]}$ for both synthetic and human noise. From this data, we can obtain the two sets of $l_2$ distances between the transition vectors – the human-synthetic distances:

$$\{d_{e,k,v}^{(1)} = ||\mathbf{p}_{e,k,v}^{human} - \mathbf{p}_{e,k,v}^{synthetic}||_2^2\}_{e \in [E], k \in [K], v \in [\nu]}$$

and the synthetic-synthetic distances:

$$\{d_{e,k,v}^{(2)} = ||\mathbf{p}_{e,k,v}^{synthetic} - \mathbf{p}_{(e+1)\%E,k,v}^{synthetic}||_2^2\}_{e \in [E], k \in [K], v \in [\nu]}.$$

From here on, we use the same formulation and procedures for the hypothesis test as Wei et al. (2022b).

## 3.3 Reproducing Noise Memorization

To further investigate the differences between artificial and real-world noise, Wei et al. (2022b), inspect the learning behavior of a classifier trained on both sets of noisy labels. Specifically, they compare the learning behavior on *Aggregate*, *Random1*, and *Worst* sets of real-world labels with their synthetic counterparts. The procedure to obtain synthetic labels remains the same as in Section 3.2. They check whether the ratio of memorized examples increases faster for synthetic or real-world noisy labels. In a $K$-class classification task, given a classifier $f$, a feature $x$ and confidence threshold $\eta$, $x$ is memorized by $f$ if $\exists i \in [K]$ s.t. $P(f(x) = i) > \eta$.

The authors do not explain the training setup beyond the fact that ResNet-34 (He et al., 2015) is used as a classifier with the confidence threshold $\eta = 0.95$. From the figures in the original paper, we deduce that each experiment was run for 150 epochs. Furthermore, we assume that a smooth learning rate schedule was used, based on the shape of the memorization curves. We obtain the rest of the hyperparameters from the memorization experiments by Xie et al. (2021) using the SGD optimizer without momentum and setting the initial learning rate to $\gamma = 1$, weight decay to $1e - 4$ and a smooth version of their learning rate scheduler $(\gamma_t = \gamma \cdot 0.1^{\frac{t}{50}})$.

### 3.4 Reproducing the Benchmark

Due to the large computation cost (see Appendix E), we select a subset of 10 methods that show good performance in the original results: *CE* (baseline), *Co-Teaching* (Han et al., 2018), *Co-Teaching+* (Yu et al., 2019), *ELR, ELR+* (Liu et al., 2020), *Divide-Mix* (Li et al., 2020), *VolMinNet* (Li et al., 2021), *CAL* (Zhu et al., 2021), *PES (semi)* (Bai et al., 2021), *SOP*, and *SOP+* (Liu et al., 2022). Here, we note that the author's reported performance for the SOP method is most likely SOP+, so we include both methods in our benchmark. To ensure a fair comparison with state-of-the-art methods we also include two newer methods: ProMix(Xiao et al., 2023) and DISC (Li et al., 2023).

The authors do not describe the evaluation procedure for the benchmark, but we have reproduced it based on GitHub issues and personal correspondence. During training with each LNL method, the classifier is evaluated in terms of micro-averaged accuracy on a clean test set every epoch. At the end, the highest achieved accuracy is reported. Each run is repeated three times to obtain the mean accuracy and the standard deviation.

While being a crucial part of many learning strategies, the benchmark authors do not discuss the hyperparameters for each method in full detail. Therefore, we reference their official implementations as a starting point and, as described by the authors, fix the following[1]: training time to 100 epochs, a stochastic gradient descent optimizer with momentum (0.9), weight decay ($5e-4$), an initial learning rate of 0.1 which decreases to 0.01 at the 60th epoch, and a minibatch size of 128. The authors also note that the ELR (+) and DivideMix methods received special treatment and were run using their original configurations. We observe that for SOP the pre-activation ResNet-18 was used [2], and based on the authors' discussion with reviewers, that the low noise configuration ($\lambda_u = 0$) was used for DivideMix [3]. Following these findings, we use ResNet-34 for most of the learning strategies and a pre-activation ResNet-18 for SOP, ELR+, and DivideMix, consistent with the described setup.

As we later show in Section 4.3, following the methodology inferred from the original study, we do not reach the same results. We argue that fixing the learning rate, optimizer, and the number of epochs without redoing the hyperparameter search reduces the performance of methods (in a possibly uneven way). Instead, when using the original hyperparameter configurations for each LNL method along with the optimizers, learning rates, and training durations, most of the methods perform close to the original reported accuracy.

Given these findings, it is likely that the original authors followed a methodology closer to our reverse-engineered approach rather than the one explicitly described in their paper. However, due to inconsistencies between the paper and the available implementation, as well as the discontinued correspondence with the original authors (see Appendix D.1), we are unable to verify this assumption with certainty.

### 3.5 Making the Benchmark Fair

We argue that this approach is flawed in several aspects and present a better evaluation procedure. When benchmarking LNL methods, the authors use a clean test set to measure accuracy every epoch, selecting the highest one at the end. This overestimates the real-world performance and is thus unreliable in practice (Cawley & Talbot, 2010).

During our experiments, we observed that the underlying classifier strongly influences the final performance. Given that the authors do not use the same ResNet model for all LNL methods (e.g., special treatment of DivideMix, ELR, and SOP), one cannot directly compare the entries on the benchmark. This is an issue as their results are currently published on a public leaderboard[4], making users unfamiliar with the testing methodology believe that the results are directly comparable between the entries.

To address these flaws, we adapt the benchmarking procedure as follows. We use the official hyperparameters for each LNL method and fix only the underlying classifier. We use the PyTorch implementation of ResNet-

---

[1]See Wei et al. (2022b) Appendix E.3.
[2]See Wei et al. (2022b) caption of Table 2 .
[3]See https://openreview.net/forum?id=TBWA6PLJZQm discussion with reviewer *EZud*.
[4]https://paperswithcode.com/task/learning-with-noisy-labels

34 with random weight initialization. Instead of testing every epoch, we use 5% of the noisy training data as a validation set to compute accuracy. We report the accuracy on the clean test set corresponding to the checkpoint with the highest validation accuracy. We present our argument on why accuracy on noisy data is a good metric for model selection in Appendix B. We repeat each run three times and report the average accuracy along with the standard deviation.

## 3.6 Experimental Setup and Code

We implement our framework using PyTorch[5] and Lightning[6] libraries for easier reproducibility, distributed training, and robust, identical testing of all methods. The modularity of the framework enables easier implementation and comparison of future LNL methods. The code is publicly available at (*URL anonymized*).

## 4 Results

In this section, we describe the results obtained using the previously specified methodology. Each subsection corresponds to one of the reproducibility claims outlined in Section 2.

## 4.1 Reproduced Noise Hypothesis Testing

We confirm the original claim that human-noisy labels in CIFAR-N differ from synthetic ones obtained from the same transition matrix. We do not match the reported p-value entirely, ours being higher at $9.5 \cdot 10^{-28}$ versus $1.8 \cdot 10^{-36}$, reported by the original authors. Qualitatively, the differences between the human and synthetic cluster transition vectors in Figure 4 are also not as prominent as in the original work. However, they still exhibit a similar pattern, with human noise varying more along the cluster dimension, which implies feature dependence. We attribute the differences to possibly different training regimes, since the original one is not described in full detail.

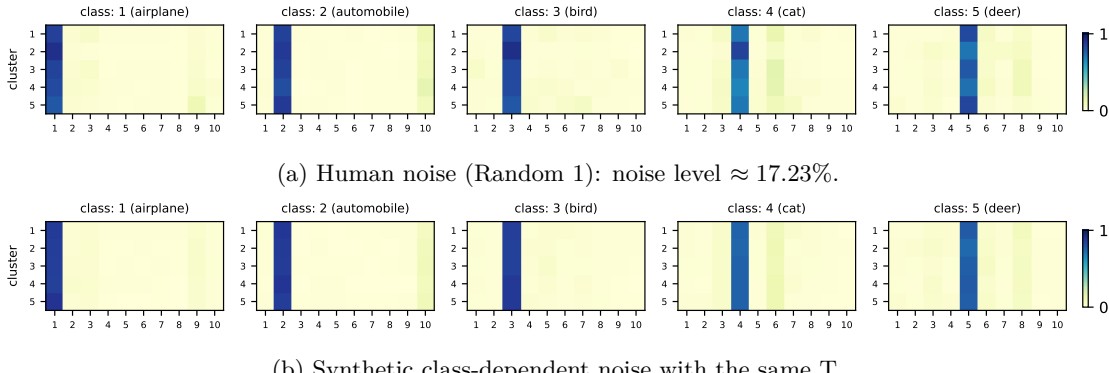

(a) Human noise (Random 1): noise level $\approx 17.23\%$.

(b) Synthetic class-dependent noise with the same T.

Figure 4: **Transition vectors** within $\nu = 5$ clusters for the first 5 classes for real-world human noise and corresponding synthetic noise. Qualitatively, human noise shows a greater diversity of transition vectors between the clusters in each class. We use the Euclidean distance for k-means clustering on $\ell^2$-normalized embeddings, which is proportional to the negative cosine similarity used by the authors.

## 4.2 Reproduced Noise Memorization

Our reimplementation of the noise memorization experiments also supports the claims of Wei et al. (2022b). Figure 5 presents the memorization dynamics across three label sets with increasing noise levels. The full lines lying above the dashed ones indicate that models begin memorizing real-world noisy labels more rapidly than synthetic ones. While the effect is less pronounced than in Figure 6 of Wei et al. (2022b), our results

---

[5]https://pytorch.org/
[6]https://lightning.ai/docs/pytorch/stable/

consistently demonstrate that models overfit human-labeled noise faster than synthetic noise, especially under higher noise levels (*Worst* label). Notably, the memorization of real-world noise remains somewhat consistent across noise levels, while synthetic labels are memorized with a higher frequency at lower noise levels. The observed discrepancies can again be attributed to differences in training configurations, as certain implementation details were either missing from the original paper or remained unclear despite our correspondence with the authors. We perform additional experiments using different types of synthetic noise and SGD parameters in Appendix C. While the memorization dynamics differ under different synthetic noises and hyperparameters, the observations of the original authors hold: deep neural nets memorize features more easily when learning with real-world human annotations than synthetic ones.

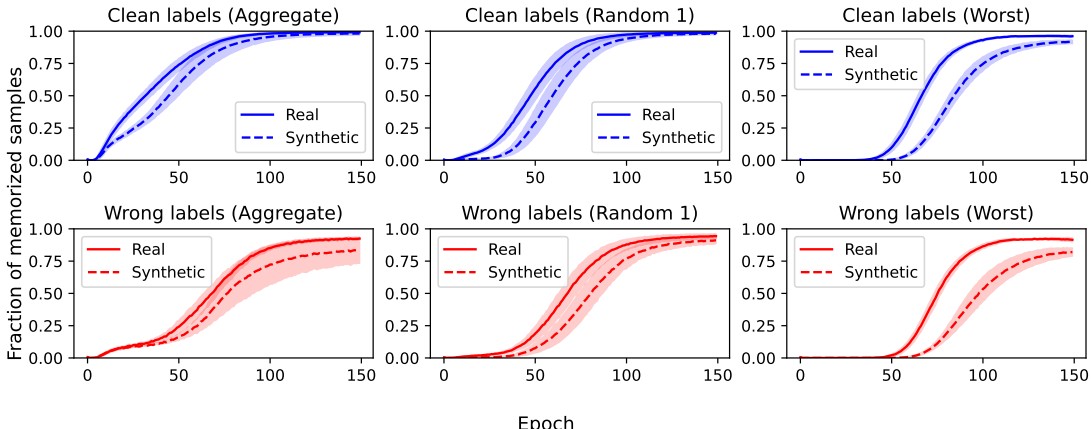

Figure 5: **Noise memorization effects**. The red line denotes the proportion of memorized (wrongly predicted) samples, blue line denotes that of correctly predicted ones. The models start to memorize the human noisy labels (full line) faster than the synthetic ones (dashed line).

## 4.3 Reproduced Benchmark

Table 1 shows the results obtained using the evaluation methodology described in Wei et al. (2022b) on the *Aggregate* label set. There are significant differences between the results reported in the original paper and those obtained when following the methodology fully (fixing the same optimizer, learning rate schedule, and number of epochs for all methods). Upon further investigation, we repeat the experiments using the methods' original configurations. The results are closer to those reported by Wei et al. (2022b).

Table 1: **Comparison with the reported results on the *Aggregate* label.** We report the results (accuracy±std) obtained using our reimplementation of the configuration described by the authors, the original configuration for each method, and the reported results. Only the results for which the authors use original configurations (denoted by *) are close to the reported ones.

| Method | Described Config | Original Config | Reported |
|---|---|---|---|
| CE | $91.70 \pm 0.07$ | $91.70 \pm 0.07$ | $87.77 \pm 0.38$ |
| Co-teaching | $90.25 \pm 0.13$ | $91.93 \pm 0.25$ | $91.20 \pm 0.13$ |
| Co-teaching+ | $86.20 \pm 0.88$ | $91.15 \pm 0.08$ | $90.61 \pm 0.22$ |
| ELR* | $93.00 \pm 0.19$ | $93.00 \pm 0.19$ | $92.38 \pm 0.64$ |
| ELR+* | $95.32 \pm 0.06$ | $95.32 \pm 0.06$ | $94.83 \pm 0.10$ |
| DivideMix* | $95.62 \pm 0.09$ | $95.62 \pm 0.09$ | $95.01 \pm 0.71$ |
| VolMinNet | $84.02 \pm 6.61$ | $90.47 \pm 0.17$ | $89.70 \pm 0.21$ |
| CAL | $91.76 \pm 0.22$ | $91.84 \pm 0.23$ | $91.97 \pm 0.32$ |
| PES (semi) | $91.53 \pm 0.44$ | $94.64 \pm 0.05$ | $94.66 \pm 0.18$ |
| SOP+ | $93.17 \pm 0.66$ | $96.04 \pm 0.15$ | $95.61 \pm 0.13$ |

We reproduce the authors' results on the CIFAR-N dataset using the configurations specific to each evaluated method. We compare our results with those reported by Wei et al. (2022b) (Tables 2 and 8). Table 2 shows the results of the reimplemented methods for all label sets in CIFAR-N.

Table 2: **Results based on the original methods' configurations.** Results for most of the methods match those reported in the original work. We report the average best test set accuracy and standard deviation of three runs.

| Method | CIFAR-10N | | | | | | CIFAR-100N | |
| | Clean | Aggregate | Random 1 | Random 2 | Random 3 | Worst | Clean | Noisy |
|---|---|---|---|---|---|---|---|---|
| CE | $94.21 \pm 0.12$ | $91.70 \pm 0.07$ | $90.20 \pm 0.04$ | $90.12 \pm 0.13$ | $90.08 \pm 0.05$ | $83.91 \pm 0.08$ | $76.23 \pm 0.19$ | $61.19 \pm 0.51$ |
| Co-teaching | $92.15 \pm 0.11$ | $91.93 \pm 0.25$ | $90.69 \pm 0.07$ | $90.51 \pm 0.14$ | $90.56 \pm 0.22$ | $80.92 \pm 0.43$ | $72.24 \pm 0.44$ | $54.48 \pm 0.27$ |
| Co-teaching+ | $92.90 \pm 0.22$ | $91.15 \pm 0.08$ | $89.82 \pm 0.13$ | $89.64 \pm 0.19$ | $89.77 \pm 0.26$ | $82.36 \pm 0.04$ | $70.39 \pm 0.45$ | $55.46 \pm 0.34$ |
| ELR | $93.97 \pm 0.12$ | $93.00 \pm 0.19$ | $92.20 \pm 0.10$ | $92.05 \pm 0.12$ | $92.18 \pm 0.17$ | $87.89 \pm 0.14$ | $75.64 \pm 0.21$ | $63.72 \pm 0.38$ |
| ELR+ | $95.81 \pm 0.16$ | $95.32 \pm 0.06$ | $94.89 \pm 0.11$ | $94.88 \pm 0.08$ | $94.93 \pm 0.07$ | $91.75 \pm 0.06$ | $\mathbf{78.82 \pm 0.24}$ | $67.87 \pm 0.07$ |
| DivideMix | $95.51 \pm 0.00$ | $95.62 \pm 0.09$ | $\mathbf{95.72 \pm 0.11}$ | $\mathbf{95.78 \pm 0.10}$ | $95.71 \pm 0.09$ | $\mathbf{93.10 \pm 0.10}$ | $78.22 \pm 0.06$ | $\mathbf{70.91 \pm 0.09}$ |
| VolMinNet | $92.71 \pm 0.02$ | $90.47 \pm 0.17$ | $88.90 \pm 0.51$ | $88.81 \pm 0.17$ | $88.67 \pm 0.10$ | $80.87 \pm 0.25$ | $72.73 \pm 0.65$ | $58.30 \pm 0.05$ |
| CAL | $93.78 \pm 0.18$ | $91.84 \pm 0.23$ | $91.10 \pm 0.26$ | $90.60 \pm 0.10$ | $90.61 \pm 0.12$ | $84.82 \pm 0.23$ | $74.53 \pm 0.21$ | $60.13 \pm 0.33$ |
| PES (semi) | $94.75 \pm 0.16$ | $94.64 \pm 0.05$ | $95.20 \pm 0.08$ | $95.26 \pm 0.13$ | $95.20 \pm 0.11$ | $92.58 \pm 0.05$ | $77.77 \pm 0.33$ | $70.32 \pm 0.28$ |
| SOP+ | $\mathbf{96.53 \pm 0.05}$ | $\mathbf{96.04 \pm 0.15}$ | $95.70 \pm 0.07$ | $95.62 \pm 0.18$ | $\mathbf{95.75 \pm 0.14}$ | $92.89 \pm 0.09$ | $77.90 \pm 0.29$ | $63.88 \pm 0.32$ |
| SOP | $93.76 \pm 0.26$ | $91.69 \pm 0.76$ | $89.83 \pm 0.24$ | $90.57 \pm 0.46$ | $90.88 \pm 0.15$ | $83.66 \pm 0.64$ | $72.97 \pm 1.15$ | $56.17 \pm 1.02$ |

For most of the methods, we are able to reproduce the results reported by the authors. One notable difference is that the baseline CE method performs much better in our experiments. The discrepancy between our and the reported baseline results is investigated in Appendix A. Some methods use noise rate as an input parameter (Han et al., 2018; Yu et al., 2019; Li et al., 2020). The parameter is fixed to the default value obtained from the original implementations. The justification for such a decision is that one does not always know how many labels are noisy in a real-world scenario. This way, the experimental protocol remains consistent, and the comparison is fair to all the methods. The decision to fix the noise level parameter could explain the difference in the Co-teaching(+) methods' performance on the CIFAR-100N noisy label set. Similarly, in the case of SOP(+) methods, symmetric noise type was assumed as an input parameter, which might explain the gap in performance on CIFAR-100N.

### 4.3.1 Human Noise vs. Synthetic Noise

Following the protocol described by the original authors, we rerun the experiment with synthetic noisy labels and compare the differences in accuracy for each entry. We subtract the real accuracies from the newly obtained synthetic ones and report the difference in Table 3.

Table 3: **Differences between the real-world and synthetic test accuracies: $\mathbf{acc_{syn} - acc_{real}}$.** Negative gaps representing the method performed better when training on human noise are highlighted in red. For CIFAR-10N, most of the methods perform better on synthetic noise, indicating a harder learning task when learning from human-assigned labels. For CIFAR-100N, this is not the case. We report the expected difference in best test set accuracy and standard deviation.

| Method | CIFAR-10N | | | | | CIFAR-100N |
| | Aggregate | Random 1 | Random 2 | Random 3 | Worst | Noisy |
|---|---|---|---|---|---|---|
| CE | $0.71 \pm 0.10$ | $0.93 \pm 0.39$ | $1.01 \pm 0.26$ | $1.10 \pm 0.05$ | $3.06 \pm 0.21$ | $1.89 \pm 0.51$ |
| Co-teaching | $0.89 \pm 0.27$ | $0.46 \pm 0.08$ | $0.09 \pm 0.15$ | $0.34 \pm 0.23$ | $1.40 \pm 0.85$ | $-2.26 \pm 0.36$ |
| Co-teaching+ | $1.05 \pm 0.16$ | $1.24 \pm 0.14$ | $1.12 \pm 0.19$ | $1.19 \pm 0.34$ | $2.92 \pm 0.31$ | $-1.94 \pm 1.10$ |
| ELR | $0.12 \pm 0.21$ | $0.31 \pm 0.18$ | $0.50 \pm 0.13$ | $0.31 \pm 0.18$ | $-3.82 \pm 0.91$ | $-0.71 \pm 0.49$ |
| ELR+ | $0.20 \pm 0.08$ | $0.31 \pm 0.13$ | $0.32 \pm 0.08$ | $0.33 \pm 0.08$ | $-0.66 \pm 2.48$ | $-0.28 \pm 0.10$ |
| DivideMix* | $0.56 \pm 0.10$ | $0.47 \pm 0.12$ | $0.54 \pm 0.10$ | $0.44 \pm 0.11$ | $1.94 \pm 0.11$ | $1.08 \pm 0.48$ |
| VolMinNet | $0.89 \pm 0.17$ | $1.15 \pm 0.51$ | $1.14 \pm 0.18$ | $1.41 \pm 0.11$ | $3.68 \pm 0.67$ | $3.11 \pm 0.15$ |
| CAL | $-0.09 \pm 0.37$ | $-0.76 \pm 0.31$ | $-0.23 \pm 0.36$ | $-0.25 \pm 0.19$ | $-0.34 \pm 0.23$ | $-0.57 \pm 0.33$ |
| PES (semi) | $0.29 \pm 0.30$ | $0.31 \pm 0.10$ | $0.41 \pm 0.23$ | $0.39 \pm 0.11$ | $1.71 \pm 0.26$ | $0.09 \pm 0.77$ |
| SOP+ | $0.25 \pm 0.16$ | $0.38 \pm 0.08$ | $0.48 \pm 0.18$ | $0.45 \pm 0.15$ | $2.07 \pm 0.09$ | $0.89 \pm 0.36$ |
| SOP | $0.70 \pm 0.99$ | $1.20 \pm 0.83$ | $0.17 \pm 0.46$ | $0.86 \pm 0.29$ | $2.73 \pm 1.29$ | $2.44 \pm 1.02$ |

We can see that, for the most part, LNL methods perform better on synthetic data. This is at least the case for CIFAR-10N, while for CIFAR-100N, the results are evenly split, with 5 methods performing better and 5 methods performing worse. This matches observations reported in Wei et al. (2022b), indicating that learning on real-world noise is more difficult to handle.

### 4.4 Fair LNL Benchmark

We now report the method performance using our benchmark as described in Section 3.5. In our evaluation, we separate the methods that train two classifiers to make a fairer comparison. In this case, we follow the original implementations and use both models and average their final predictions when calculating the accuracies. The results are summarized in Table 4. We report the test set accuracy on the clean test for the checkpoint that achieved the highest accuracy on the noisy validation set. A similar approach to be used in practice was suggested by Jiang et al. (2020). For details on our choice, please refer to Appendix B.

Looking at the results in Table 4, we first observe that all methods perform worse than in Table 2. This is because we use the PyTorch ResNet-34 implementation here instead of the modified one used by the authors (more on that in Appendix F). Otherwise, the model ranking remains similar, with DivideMix (Li et al., 2020) and SOP+ (Liu et al., 2022) performing best from the methods used in the original work. The state-of-the-art method ProMix consistently outperforms all methods on CIFAR-10N. However, on CIFAR-100N, DivideMix retains the top ranking. Among the methods that use a single model, SOP+ outperforms the newer DISC method on CIFAR-10N, while the latter achieves the best performance on CIFAR-100N. We still observe that baseline CE training performs better than previously reported, consistently beating CAL (Zhu et al., 2021), VolMinNet (Li et al., 2021), and SOP (Liu et al., 2022).

Table 4: **Fair LNL Benchmark**. To ensure a fair comparison, all methods utilize a ResNet-34 backbone and are evaluated using their original hyperparameter configurations. We report the test accuracy for the model checkpoint that achieves the highest validation accuracy on noisy labels. Since some methods train two classification models, we identify the best-performing method within each group (underlined) and the overall best-performing method (**bolded**). The top group utilizes a single model, and the bottom group two. For all methods, we report the mean and standard deviation across three runs.

| Method | CIFAR-10N | | | | | | CIFAR-100N | |
|---|---|---|---|---|---|---|---|---|
| | Clean | Aggregate | Random 1 | Random 2 | Random 3 | Worst | Clean | Noisy |
| CE | $85.50 \pm 0.25$ | $83.50 \pm 0.23$ | $81.96 \pm 0.77$ | $82.00 \pm 0.13$ | $82.19 \pm 0.31$ | $75.48 \pm 0.30$ | $59.18 \pm 0.18$ | $49.51 \pm 0.39$ |
| ELR+ | $88.56 \pm 0.23$ | $87.76 \pm 0.16$ | $86.75 \pm 0.15$ | $86.94 \pm 0.07$ | $86.97 \pm 0.07$ | $81.50 \pm 0.12$ | $61.80 \pm 0.25$ | $52.91 \pm 0.27$ |
| VolMinNet | $82.57 \pm 0.20$ | $80.42 \pm 0.36$ | $79.18 \pm 0.10$ | $78.54 \pm 0.30$ | $78.56 \pm 0.13$ | $72.07 \pm 0.49$ | $51.27 \pm 0.18$ | $42.39 \pm 0.74$ |
| CAL | $83.38 \pm 0.47$ | $81.51 \pm 0.21$ | $79.64 \pm 0.45$ | $79.50 \pm 0.31$ | $79.67 \pm 0.26$ | $73.08 \pm 0.88$ | $58.02 \pm 0.17$ | $47.37 \pm 0.49$ |
| PES (semi) | $87.08 \pm 0.08$ | $86.62 \pm 0.23$ | $87.63 \pm 0.11$ | $87.47 \pm 0.20$ | $86.80 \pm 0.66$ | $84.12 \pm 0.34$ | $60.51 \pm 0.14$ | $52.95 \pm 0.47$ |
| SOP | $82.12 \pm 0.43$ | $79.78 \pm 0.85$ | $79.22 \pm 0.57$ | $79.37 \pm 0.29$ | $78.66 \pm 0.57$ | $71.76 \pm 1.98$ | $52.61 \pm 0.43$ | $40.98 \pm 0.28$ |
| SOP+ | $\underline{90.09 \pm 0.03}$ | $\underline{89.50 \pm 0.11}$ | $\underline{88.82 \pm 0.43}$ | $\underline{88.99 \pm 0.16}$ | $\underline{88.72 \pm 0.19}$ | $\underline{85.18 \pm 0.59}$ | $61.27 \pm 0.39$ | $50.85 \pm 0.06$ |
| DISC | $89.12 \pm 0.12$ | $88.12 \pm 0.24$ | $87.31 \pm 0.28$ | $87.55 \pm 0.25$ | $87.44 \pm 0.33$ | $83.04 \pm 0.13$ | $\underline{62.13 \pm 0.37}$ | $\underline{53.09 \pm 0.47}$ |
| Co-teaching | $86.56 \pm 0.08$ | $86.56 \pm 0.27$ | $85.26 \pm 0.15$ | $85.11 \pm 0.27$ | $84.56 \pm 0.22$ | $75.62 \pm 0.32$ | $57.07 \pm 0.16$ | $43.26 \pm 0.43$ |
| Co-teaching+ | $86.59 \pm 0.12$ | $85.16 \pm 0.32$ | $83.71 \pm 0.28$ | $84.20 \pm 0.21$ | $83.92 \pm 0.15$ | $76.71 \pm 0.50$ | $54.76 \pm 0.58$ | $44.25 \pm 0.34$ |
| ELR+ | $88.56 \pm 0.23$ | $87.76 \pm 0.16$ | $86.75 \pm 0.15$ | $86.94 \pm 0.07$ | $86.97 \pm 0.07$ | $81.50 \pm 0.12$ | $61.80 \pm 0.25$ | $52.91 \pm 0.27$ |
| DivideMix | $89.59 \pm 0.12$ | $89.36 \pm 0.22$ | $89.34 \pm 0.10$ | $89.61 \pm 0.10$ | $89.28 \pm 0.31$ | $86.82 \pm 0.17$ | $\mathbf{64.21 \pm 0.02}$ | $\mathbf{56.21 \pm 0.38}$ |
| ProMix | $\mathbf{91.53 \pm 0.26}$ | $\mathbf{90.72 \pm 0.14}$ | $\mathbf{90.57 \pm 0.20}$ | $\mathbf{90.61 \pm 0.13}$ | $\mathbf{90.61 \pm 0.25}$ | $\mathbf{88.99 \pm 0.14}$ | $63.53 \pm 0.03$ | $55.67 \pm 0.10$ |

## 5 Discussion

We managed to reproduce the hypothesis testing and noise memorization effects. Due to a partially unclear description of the experiment protocol, we do not obtain the same results as the authors, but the results nevertheless lead to the same conclusions. The real-world label noise is indeed different from its synthetic counterpart, and the classifiers start to overfit on it faster, which indicates a harder learning task.

While the authors describe the benchmarking experiment in some detail, we fail to reach the same results using their methodology. Instead, when we use the original hyperparameter configurations for each LNL method, we obtain results closer to the reported accuracy for most of the methods. Perhaps most interesting is that the baseline cross-entropy training performs significantly better than expected, outperforming several

methods designed explicitly for LNL scenarios even in the most difficult noise settings. This behavior also persists outside our framework, even in the original code provided by the authors, as we show in Appendix A. This finding leaves an avenue for possible future work. We provide additional comments on reproducibility and our discussion with the authors in Appendix D.

### 5.1   Additional Observations

During our implementation of several LNL methods, we noticed that the original implementations incorrectly generate synthetic noise. This raises the question of their reported performances. The symmetric noise produced by the official DivideMix (Li et al., 2020) implementation [7] uses a biased noise rate. For a given noise rate $r$, their implementation randomly selects a fraction $r$ of clean labels. The noisy labels are sampled uniformly from all classes $k$. This results in $\frac{1}{k}$ of the samples agreeing with the original label, resulting in the actual noise rate being $\widetilde{r} = r - \frac{1}{k}r$ instead of $r$. This is especially evident in high-noise scenarios where DivideMix performs best. We observe an even bigger error in the ELR (Liu et al., 2020) implementation of synthetic noise. The method uses two models trained on separate datasets, which are both noised independently, while also producing noisy labels with a decreased noise rate $\widetilde{r}$ described above [8]. Here, without loss of generality, we treat the sample as noisy and as having the wrong label in the first dataset. This results in the probability of at least one of the models observing the correct label for a sample given that the sample is noisy, being $1 - \widetilde{r}$ instead of 0.

This brings into question the results reported in the respective original works. However, for our benchmark reproduction, where human noise is used and synthetic noise pipelines are unified, the results are unaffected, and both methods rank among the top-performing ones.

### 5.2   Conclusion

This reproducibility study examined the claims made by  Wei et al. (2022b) regarding real-world label noise. We successfully replicated the original noise hypothesis testing and noise memorization experiments with minor quantitative discrepancies likely stemming from undisclosed training details. Despite these discrepancies, our results support the authors' claim that learning on datasets with real-world noisy labels is more challenging than with synthetic noise. However, our attempts to reproduce the benchmark results revealed significant deviations. Through a detailed review and reverse engineering of the original codebase, we identified inconsistencies in the paper's description, particularly regarding hyperparameter configurations. We then developed a unified framework for LNL methods, accompanied by an improved benchmarking procedure, incorporating a validation set on noisy data and consistent use of the ResNet-34 architecture. This framework provides a more robust platform for evaluating LNL methods, especially in scenarios where clean validation data is unavailable. Our findings highlight the need for comprehensive documentation and transparency in scientific research to ensure reproducibility and foster progress in the field.

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

## A    Baseline Performance Results

The largest discrepancy between our and the authors' results comes from the baseline (CE) training. Wei et al. (2022b), do not describe the exact procedure for obtaining their results. In our correspondence (see Appendix D.1), they clarify that they used the best test set performance and report the mean and standard deviation across five random seeds. This procedure results in a discrepancy between our reproduced results and the ones reported in the original work (Wei et al., 2022b). We visualize the discrepancy in Figure 6.

We can see that the accuracy of the best checkpoint in our experiments is higher than their reported performance across all runs. We were not alone in noticing this discrepancy, as at the time of investigation, there were two open issues on the authors' official Github repository with the same question. Authors of the issues report the last obtained accuracies on the *Worst* label set:≈ 68% [9] and ≈ 67% [10], which are also in line with our last epoch accuracy ($68 \pm 0.88$).

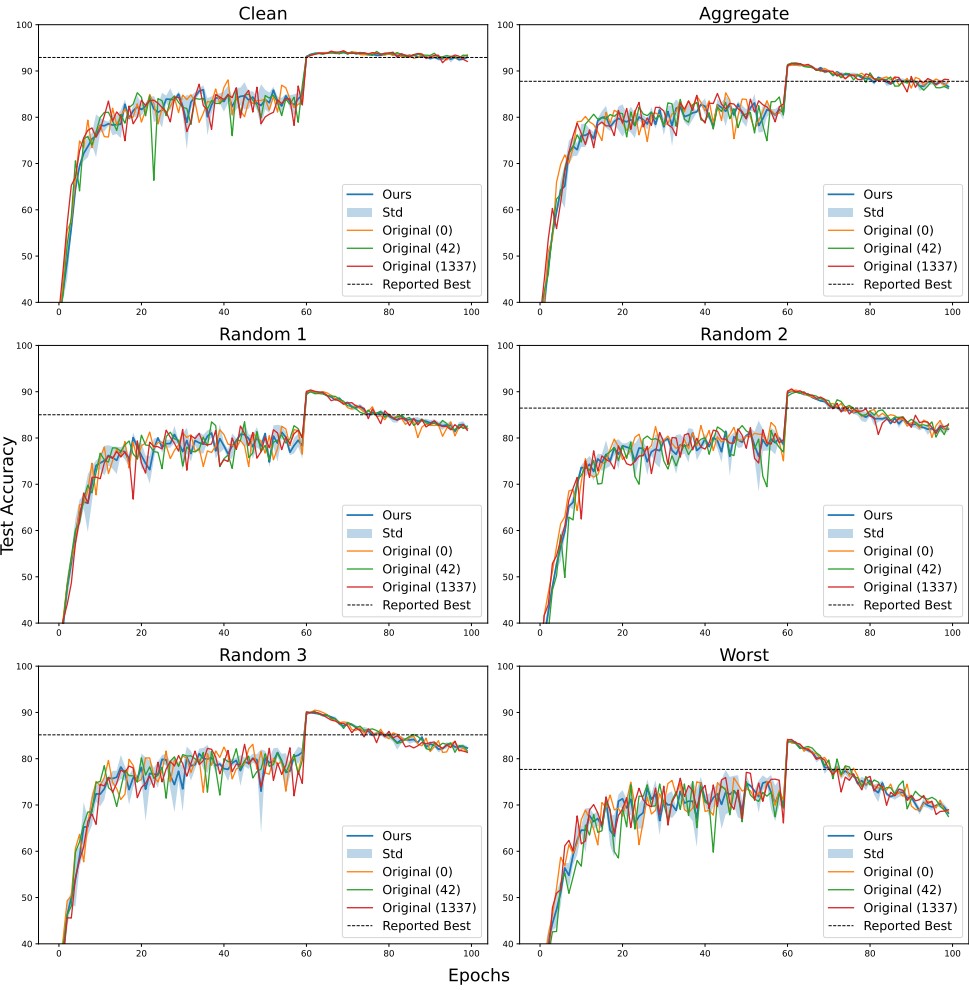

Figure 6: **Differences between the authors' reported (dotted) baseline performance and accuracy obtained in reproduction (full blue) as well as using the original code (orange, red, yellow).** The performance obtained using a reimplemented framework perfectly aligns with the original code. However, on all the noise labels, their reported accuracy does not coincide with the best accuracy for any of the runs.

---

[9]`https://github.com/UCSC-REAL/cifar-10-100n/issues/5`
[10]`https://github.com/UCSC-REAL/cifar-10-100n/issues/8`

## B Argument For Using Accuracy On Noisy Data

Since testing every epoch and reporting the highest result overestimates performance and is thus considered a bad practice, we have to find an alternative. We want to use the established train-validation-test split in this case. The first problem in LNL setting arises when we need to decide whether the validation set is noisy or not, since practices for this are not yet established in the field. Given the nature of the task, we argue that the validation set should be noisy as well (Jiang et al., 2020). If one has non-noisy data, one can train on it, implicitly changing the noise ratio of the data. Additionally, some LNL methods (Patrini et al., 2017; Wei & Liu, 2021) use validation data to estimate parameters for loss functions that are later used to train. If validation data were to be clean, there would be a direct data leak in such cases.

Consequently, we must decide on a proxy metric for the accuracy on clean data. We consider two options: using validation losses of each LNL method or validation accuracy. Given that losses vary between the LNL methods (some have no upper bounds) and are not as stable (see Figure 7), we propose to use accuracy on noisy validation data for model selection. We propose this, as it remains stable across all tested methods and strongly correlates with final test performance, preserving model ranking with high probability.

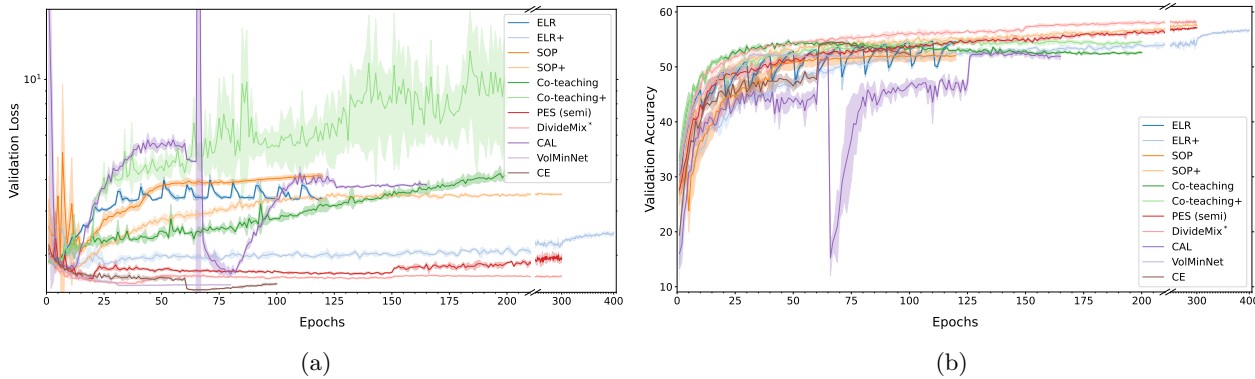

(a)             (b)

Figure 7: **Validation performance on noisy labels** from our benchmark implementations on the *Worst* label set. Some methods may assign near-zero probabilities to noisy labels, and due to the unbounded nature of the cross-entropy loss, this leads to high and unstable loss values (a). However, noisy validation accuracy remains stable across runs and serves as a reliable predictor of clean test-set performance (b).

We verify our claims empirically by testing how well our approach ranks the models in comparison to the rankings of their best test set accuracies. For each method, we select the best performance on the test data and the test performance corresponding to the best validation checkpoint. We average the results across three runs and compute the Kendall rank correlation coefficient (Kendall, 1938) for method rankings. For all label sets we obtain statistically significant results, indicating a strong relationship between the ordering based on noisy validation accuracy and clean test set performance. We report all the orderings with their corresponding p-value and Kendall-$\tau$ statistics below, where we underline all changes in rank:

> CIFAR 100 - *Clean* ($\tau = 1.00$, $p = 5.01e - 08$)
>
> Val: DivideMix, ELR+, SOP+, PES semi, CE, ELR, CAL, Co-Te., Co-Te.+, SOP, VolMinNet
>
> Test: DivideMix, ELR+, SOP+, PES semi, CE, ELR, CAL, Co-Te., Co-Te.+, SOP, VolMinNet
>
> CIFAR 100N - *Noisy* ($\tau = 1.00$, $p = 5.01e - 08$)
>
> Val: DivideMix, PES semi, ELR+, SOP+, CE, ELR, CAL, Co-Te.+, Co-Te., VolMinNet, SOP
>
> Test: DivideMix, PES semi, ELR+, SOP+, CE, ELR, CAL, Co-Te.+, Co-Te., VolMinNet, SOP

CIFAR 10 - *Clean* ($\tau = 1.0$, $p = 5.01e - 08$)

Val: SOP+, DivideMix, ELR+, PES semi, Co-Te.+, Co-Te., CE, ELR, CAL, VolMinNet, SOP

Test: SOP+, DivideMix, ELR+, PES semi, Co-Te.+, Co-Te., CE, ELR, CAL, VolMinNet, SOP

CIFAR 10N - *Aggregate* ($\tau = 0.82$, $p = 1.32e - 04$)

Val: SOP+, DivideMix, ELR+, PES semi, Co-Te, Co-Te.+, ELR, CE, CAL, VolMinNet, SOP

Test: SOP+, DivideMix, ELR+, Co-Te., PES semi, Co-Te.+, ELR, CE, CAL, VolMinNet, SOP

CIFAR 10N - *Random 1* ($\tau = 0.6$, $p = 9.95e - 03$)

Val: DivideMix, SOP+, PES semi, ELR+, Co-Te., ELR, Co-Te.+, CE, CAL, SOP, VolMinNet

Test: DivideMix, SOP+, PES semi, ELR+, Co-Te., Co-Te.+, ELR, CE, CAL, SOP, VolMinNet

CIFAR 10N - *Random 2* ($\tau = 1.0$, $p = 5.01e - 08$)

Val: DivideMix, SOP+, PES semi, ELR+, Co-Te., Co-Te.+, ELR, CE, CAL, SOP, VolMinNet

Test: DivideMix, SOP+, PES semi, ELR+, Co-Te., Co-Te.+, ELR, CE, CAL, SOP, VolMinNet

CIFAR 10N - *Random 3* ($\tau = 0.64$, $p = 5.71e - 03$)

Val: DivideMix, SOP+, ELR+, PES semi, Co-Te., Co-Te.+, ELR, CE, CAL, SOP, VolMinNet

Test: DivideMix, SOP+, PES semi, ELR+, Co-Te., Co-Te.+, ELR, CE, CAL, VolMinNet, SOP

CIFAR 10N - *Worst* ($\tau = 1.0$, $p = 5.01e - 08$)

Val: DivideMix, SOP+, PES semi, ELR+, ELR, Co-Te.+, Co-Te., CE, CAL, VolMinNet, SOP

Test: DivideMix, SOP+, PES semi, ELR+, ELR, Co-Te.+, Co-Te., CE, CAL, VolMinNet, SOP

Yuan et al. (2024) propose LabelWave, a model selection criterion for training with noisy labels that eliminates the need for a validation set. We consider LabelWave a promising alternative to our current model selection strategy based on noisy validation accuracy. Benchmarking existing LNL methods using LabelWave as the selection criterion represents an interesting and valuable direction for future work.

## C   Additional Memorization Experiments

We extend the memorization experiments to include a broader range of synthetic noise types and hyperparameter configurations. These experiments follow the procedure described in Section 3.3.

To evaluate the effect of different types of synthetic noise, we replace the asymmetric transition matrix used in Section 4.2 with symmetric and pair-flipping noise matrices. The transition matrices are constructed following the definitions in Han et al. (2018), and the overall noise rate $\rho$ is estimated from the corresponding real-world noisy labels.

For symmetric noise, we set the diagonal elements of the transition matrix to $1 - \rho$ and all off-diagonal elements to $\frac{\rho}{C-1}$, where $C$ is the number of classes. For pair-flipping noise, we again set the diagonal entries to $1 - \rho$, and assign the entire off-diagonal mass $\rho$ to the most common transition class observed in the real-world noise.

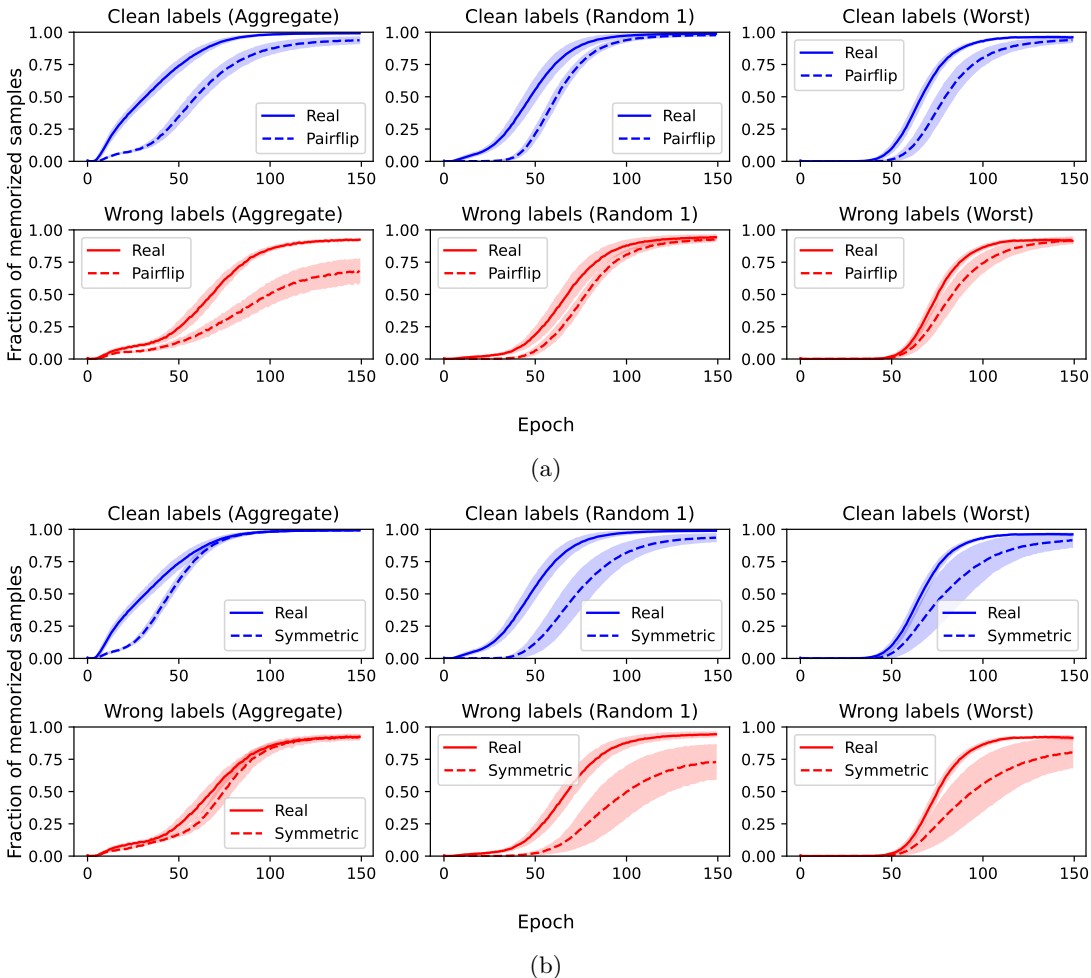

Figure 8: **Effects of different types of synthetic noise on memorization**. (a) Pair-flipping noise constructed based on the most common transitions observed in human-labeled data. (b) Symmetric noise applied uniformly across all incorrect classes. In both cases, models begin to memorize human-labeled noise earlier than synthetic noise, despite different memorization dynamics.

We present the results in Figure 8. While synthetic noise types lead to varying memorization behaviors, we observe that the models still overfit human noisy labels faster than synthetic ones.

We also investigate how different training hyperparameters influence memorization dynamics. Building on the setup described in Section 3.3, we vary one hyperparameter at a time while keeping the others fixed. In the first experiment, we replace the exponential learning rate decay with a step decay, reducing the learning rate by a factor of $\gamma = 0.1$ at epoch 50. In the second experiment, we increase the initial learning rate to 0.1. In the third, we raise the weight decay to $5e - 4$.

The results, shown in Figure 9, indicate that while these hyperparameter changes do affect memorization behavior, the model still consistently memorizes real-world human label noise faster than synthetic noise. This further supports our conclusion that human-labeled, instance-dependent noise is inherently more susceptible to early memorization than instance-independent synthetic noise.

# D  Comments on Reproducibility

In this section, we document our efforts to reproduce the key experiments by Wei et al. (2022b), highlighting both successes and challenges. Table 5 summarizes the classification of reproducibility outcomes, indicating

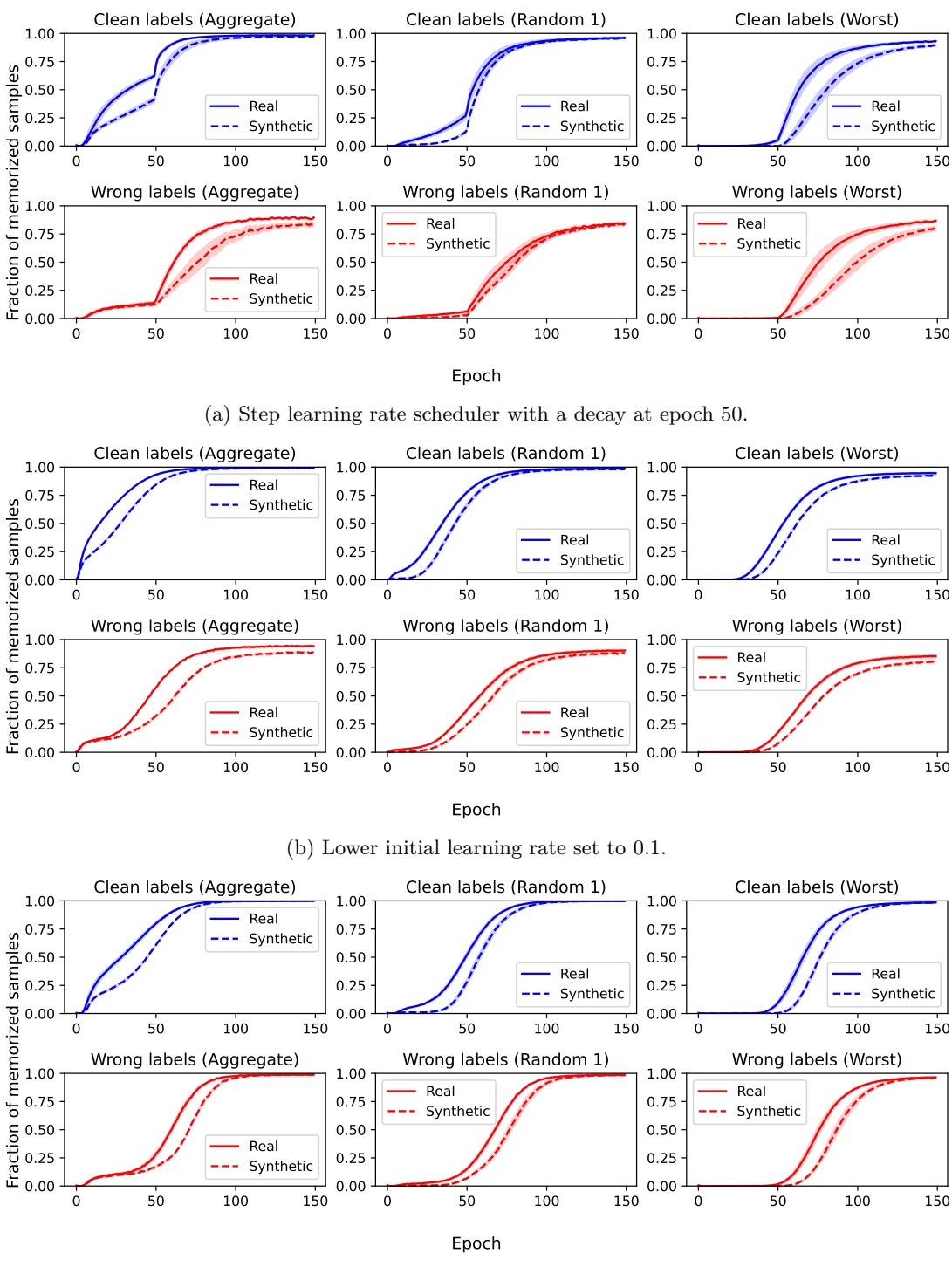

(a) Step learning rate scheduler with a decay at epoch 50.

(b) Lower initial learning rate set to 0.1.

(c) Increased weight decay set to $5e-4$.

Figure 9: **Effect of hyperparameters on memorization.** We observe different memorization dynamics across hyperparameter setups. However, models consistently memorize human-labeled noise earlier than synthetic noise

where discrepancies arose. We also detail our communication with the original authors, the aspects of

the reproduction process that were straightforward, and the challenges we faced, particularly regarding undocumented hyperparameters, inconsistent training protocols, and unclear benchmarking procedures.

Table 5: **Classification of reproducibility results.**

| Experiment | Discrepancy | Result |
|---|---|---|
| Noise hypothesis testing | Missing details in experimental setup. | Reproduced |
| Memorization effects | Missing details in experimental setup. | Reproduced |
| Baseline performance | Difference in reported vs. observed results. | Different |
| LNL Benchmark | Ambiguities in paper descriptions. | Reimplemented |

### D.1 Communication with Original Authors

We contacted the authors two times during our implementation effort. The first time, we inquired about several inconsistencies between the paper and the provided code. We also asked about the hyperparameter selection protocol, which backbone models were used, which checkpoints were selected, and some general evaluation and method-specific questions. The authors responded with a short email, answering some of our questions but avoiding our questions about inconsistencies.

After some time, we contacted them for the second time. We inquired how the validation sets were handled, as some methods could not be properly trained without them. We also asked about the specifics of the hypothesis testing setup and the differences between the original and our reproduced results. This time, the authors did not respond.

### D.2 What was easy

Using the baseline code and the datasets provided by the benchmark authors was easy and only required minimal tweaking to run. Running most of the LNL strategies' repositories was also straightforward, as most authors include instructions for basic use cases and experiment reproduction in the source code repository.

### D.3 What was difficult

Throughout the reproduction attempt, we encountered several problems, most of them stemming from unclear and ambiguous descriptions of the benchmark. Many evaluated methods rely on algorithm-specific hyperparameters, which the authors did not list. Therefore, an effort was made to try to recover them by experimentally trying out different combinations in an attempt to match the reported results. The learning rate schedule for all methods was said to follow a multi-step schedule with a single decay at the 50th epoch. In the original source code, the switch happens at the 60th epoch.

Several LNL methods use different warm-up training techniques. Some fully reset the model, while others use the warm-up epochs to find a set of weights that perform best on the validation set and later continue the training from the pre-trained checkpoint. A shared checkpoint was supposed to be used for the benchmark. However, the paper does not mention how such a checkpoint was obtained. After communication with the authors, they clarified that the warmup checkpoint was obtained by training the model with cross-entropy loss for 200 epochs.

## E Computational Requirements

We run all our experiments on a system with 8 Nvidia Titan X Pascal GPUs running Ubuntu 20.04.2 LTS and CUDA version 12.2. With this setup, it takes approximately 1.5 GPU hours to reproduce the noise hypothesis testing experiment and approximately 27 GPU hours to reproduce the noise memorization experiment. Reproducing benchmark results on the selected subset of the LNL methods takes approximately 52.5 GPU hours per label set; we present a breakdown of the training times in Table 6. It takes approximately

$2,000$ GPU hours to reproduce our results fully (eight label sets, each with three repeats, and six synthetic runs with two repeats).

Table 6: **Runtimes of the 10 selected methods**, for a single repeat of a single noise label.

| Method | Runtime (h) |
|---|---|
| CE | 1.5 |
| Co-teaching (+) | 5.6 |
| ELR | 1.8 |
| ELR+ | 7.9 |
| DivideMix | 14.0 |
| VolMinNet | 1.3 |
| CAL | 1.5 |
| PES (semi) | 12.2 |
| SOP+ | 5.5 |
| SOP | 1.2 |
| total | 52.5 |

## F  Hyperparameters

In this section, we report the hyperparameters for all the methods included in our reproduced evaluation to enable the replication of our results. Table 8 describes the hyperparameters of methods using the *Pre-ActResNet18* (He et al., 2016) backbone, and Table 9 for the methods using the *ResNet-34* (He et al., 2015) backbone. Here we note that most of the methods' original implementations as well as the model implementation provided by the authors [11] use ResNet implementations that use stride of 1 instead of 2 in the first layer, resulting in a four times increase in activation volumes. The modified implementation also excludes the first max pooling layer (see He et al. (2015) Table 1), resulting in another four-times increase in activation volumes, for a total of 16 times bigger activation volumes. Modifying these layers increases the baseline performance significantly (10% when comparing baselines in Tables 2 and 4).

In our reproduction experiments, we use this version of ResNet since it was provided in the author's original repository. For our benchmark (Section 4.4) we use the same hyperparameters as described in tables 9 and 8 with the exception of using the official PyTorch implementation of ResNet [12]. We report the parameters for newer methods DISC and ProMix in Table 7.

---

[11]https://github.com/UCSC-REAL/cifar-10-100n/blob/main/models/resnet.py
[12]https://pytorch.org/vision/main/models/resnet.html

Table 7: Hyperparameters for newer methods.

| Method | Hyperparameter | CIFAR10N | CIFAR100N |
|---|---|---|---|
| DISC | optimizer | SGD | SGD |
| | lr | 0.1 | 0.1 |
| | weight_decay | 0.001 | 0.001 |
| | SGD momentum | 0 | 0 |
| | scheduler | MultiStepLR ([80, 160], 0.1) | MultiStepLR ([80, 160], 0.1) |
| | epochs | 200 | 200 |
| | alpha | 5.0 | 5.0 |
| | start_epoch | 15 | 15 |
| | sigma | 0.5 | 0.5 |
| | momentum | 0.99 | 0.99 |
| | lambda_ce | 1.0 | 1.0 |
| | lambda_h | 1.0 | 1.0 |
| ProMix | optimizer | SGD | SGD |
| | lr | 0.05 | 0.05 |
| | weight_decay | 0.0005 | 0.0005 |
| | momentum | 0.9 | 0.9 |
| | scheduler | CosineAnnealingLR | CosineAnnealingLR |
| | epochs | 600 | 600 |
| | warmup_epochs | 10 | 30 |
| | rampup_epochs | 50 | 50 |
| | noise_type | symmetric | symmetric |
| | rho_start, rho_end | (0.5, 0.5) | (0.5, 0.5) |
| | debias_beta_pl | 0.8 | 0.5 |
| | alpha_output | 0.8 | 0.5 |
| | tau | 0.99 | 0.95 |
| | start_expand | 250 | 250 |
| | threshold | 0.9 | 0.9 |
| | bias_m | 0.9999 | 0.9999 |
| | temperature 0.5 | 0.5 | |
| | feat_dim | 128 | 128 |

Table 8: Hyperparameters for methods using PreActResNet18 backbone.

| Method | Hyperparameter | CIFAR10N | CIFAR100N |
|---|---|---|---|
| DivideMix | optimizer | SGD | SGD |
| | lr | 0.02 | 0.02 |
| | weight_decay | 0.0005 | 0.0005 |
| | momentum | 0.9 | 0.9 |
| | scheduler | LambdaLR | LambdaLR |
| | epochs | 300 | 300 |
| | alpha | 4 | 4 |
| | noise_type | asymmetric | asymmetric |
| | p_thresh | 0.5 | 0.5 |
| | temperature | 0.5 | 0.5 |
| | lambda_u | 0 | 0 |
| | warmup_epochs | 10 | 30 |
| ELR+ | optimizer | SGD | SGD |
| | lr | 0.02 | 0.02 |
| | weight_decay | 0.0005 | 0.0005 |
| | momentum | 0.9 | 0.9 |
| | scheduler | MultiStepLR ([150], 0.1) | MultiStepLR ([200], 0.1) |
| | epochs | 200 | 250 |
| | beta | 0.7 | 0.9 |
| | lmbd | 3 | 7 |
| | alpha | 1 | 1 |
| | gamma | 0.997 | 0.997 |
| | ema_step | 40000 | 40000 |
| | coef_step | 0 | 40000 |
| SOP+ | optimizer | SGD | SGD |
| | lr | 0.02 | 0.02 |
| | weight_decay | 0.0005 | 0.0005 |
| | momentum | 0.9 | 0.9 |
| | scheduler | CosineAnnealing | CosineAnnealing |
| | epochs | 300 | 300 |
| | lr_u | 10 | 1 |
| | lr_v | 10 | 10 |
| | overparam_mean | 0.0 | 0.0 |
| | overparam_std | 1e-08 | 1e-08 |
| | ratio_balance | 0.1 | 0.1 |
| | ratio_consistency | 0.9 | 0.9 |

Table 9: Hyperparameters for methods using ResNet-34 backbone.

| Method | Hyperparameter | CIFAR-10N | CIFAR-100N |
|---|---|---|---|
| CAL | optimizer | SGD | SGD |
| | lr | 0.1 | 0.1 |
| | weight_decay | 0.0005 | 0.0005 |
| | momentum | 0.9 | 0.9 |
| | scheduler | MultiStepLR ([60], 0.1) | MultiStepLR ([60], 0.1) |
| | epochs | 165 | 165 |
| | alpha | 0.0 | 0.0 |
| | alpha_scheduler | seg | seg |
| | warmup_epochs | 65 | 65 |
| | alpha_list_warmup | [0.0, 2.0] | [0.0, 1.0] |
| | milestones_warmup | [10, 40] | [10, 40] |
| | alpha_list | [0.0, 1.0, 1.0] | [0.0, 1.0, 1.0] |
| | milestones | [10, 40, 80] | [10, 40, 80] |
| CE | optimizer | SGD | SGD |
| | lr | 0.1 | 0.1 |
| | weight_decay | 0.0005 | 0.0005 |
| | momentum | 0.9 | 0.9 |
| | scheduler | MultiStepLR ([60], 0.1) | MultiStepLR ([60], 0.1) |
| | epochs | 100 | 100 |
| Co-teaching | optimizer | Adam | Adam |
| | lr | 0.001 | 0.001 |
| | weight_decay | 0 | 0 |
| | scheduler | alpha_schedule | alpha_schedule |
| | epochs | 200 | 200 |
| | forget_rate | 0.2 | 0.2 |
| | exponent | 1 | 1 |
| | num_gradual | 10 | 10 |
| | epoch_decay_start | 80 | 100 |
| Co-teaching+ | optimizer | Adam | Adam |
| | lr | 0.001 | 0.001 |
| | weight_decay | 0 | 0 |
| | scheduler | alpha_schedule | alpha_schedule |
| | epochs | 200 | 200 |
| | init_epoch | 20 | 5 |
| | forget_rate | 0.2 | 0.2 |
| | exponent | 1 | 1 |
| | num_gradual | 10 | 10 |
| | epoch_decay_start | 80 | 80 |
| ELR | optimizer | SGD | SGD |
| | lr | 0.02 | 0.02 |
| | weight_decay | 0.001 | 0.001 |
| | momentum | 0.9 | 0.9 |
| | scheduler | CosineAnnealing | MultiStepLR ([80, 120], 0.01) |
| | epochs | 120 | 150 |
| | beta | 0.7 | 0.9 |
| | lmbd | 3 | 7 |
| PES (semi) | optimizer | SGD | SGD |
| | lr | 0.02 | 0.02 |
| | weight_decay | 0.0005 | 0.0005 |
| | momentum | 0.9 | 0.9 |
| | scheduler | CosineAnnealing | CosineAnnealing |
| | epochs | 300 | 300 |
| | PES_lr | 0.0001 | 0.0001 |
| | warmup_epochs | 20 | 35 |
| | T2 | 5 | 5 |
| | lambda_u | 5 | 75 |
| | temperature | 0.5 | 0.5 |
| | alpha | 4 | 4 |
| SOP | optimizer | SGD | SGD |
| | lr | 0.02 | 0.02 |
| | weight_decay | 0.0005 | 0.0005 |
| | momentum | 0.9 | 0.9 |
| | scheduler | MultiStepLR ([40, 80], 0.1) | MultiStepLR ([40, 80], 0.1) |
| | epochs | 120 | 150 |
| | lr_u | 10 | 1 |
| | lr_v | 10 | 10 |
| | overparam_mean | 0.0 | 0.0 |
| | overparam_std | 1e-08 | 1e-08 |
| | ratio_balance | 0.0 | 0.0 |
| | ratio_consistency | 0.0 | 0.0 |
| VolMinNet | optimizer | SGD | SGD |
| | lr | 0.01 | 0.01 |
| | weight_decay | 0.0001 | 0.0001 |
| | momentum | 0.9 | 0.9 |
| | scheduler | MultiStepLR ([30, 60], 0.1) | MultiStepLR ([30, 60], 0.1) |
| | epochs | 80 | 80 |
| | lam | 0.0001 | 0.0001 |
| | init_t | 2 | 2 |
| | optimizer_transition_mtx | SGD | Adam |

