# OpenReview forum: "Learning with Noisy Labels [Re]visited"
_TMLR — Rejected by TMLR_

### Review · Reviewer_NuQu · 2025-03-18

**Summary Of Contributions:**

1) The authors reproduce the experiments of Wei et al. (2022) and share their code.
2) The authors implement a unified framework and propose an improved benchmarking protocol that helps better evaluations of LNL methods.
3) They benchmark LNL methods in a controlled environment, enabling a fair comparison of LNL methods on CIFAR-N.

**Audience:**

Yes

**Broader Impact Concerns:**

No broader impact concerns.

**Claims And Evidence:**

Yes

**Requested Changes:**

N/A

**Strengths And Weaknesses:**

I primarily conduct theoretical research in machine learning and do not specialize in LNL. I am unsure why this type of paper was assigned to me. However, from a non-expert perspective,

1) the effort to openly share existing research, verify benchmarks, and establish a more reasonable framework seems to be a meaningful research direction.
2) They provide a detailed description of what previous authors (We et al, 2022) did and how they changed or verified their work.

These seem like strengths. However, as a non-expert,

1) I am not sure how valuable this 'reproducing of previous paper' is. It seems valuable, but I'm not sure.
2) I don't fully understand why the previous work is that unfair. The only thing that I can check is that they made the codes public and checked the benchmarks. Their 'unfairness' is written only on Section 3.5, but most of the things, especially the clean test set, look tedious.


Again, since I am not an expert for this type of experiment-verification papers, I don't have a strong opinion, but I also believe keep checking reproducibility and integrity of existing papers seems valuable, so I am slightly lean towards acceptance.

---

> ### Author Response · Authors · 2025-03-24
>
> We thank the reviewer for their time and effort to review our paper.
>
> According to the reviewer, they are not an expert in LNL. However, we believe their background in theoretical ML offers a valuable perspective. LNL focuses on designing algorithms that account for noise in the data-generating process, often relying on theoretical foundations. With this in mind, we address the concerns they raised below.
>
> **1. Regarding concerns about the value of reproducibility:**
>
> Our work falls within the reproducibility scope of TMLR (reproducibility studies of previously published results or claims). It aims to both verify the claims and findings of the original work and go beyond it to improve the reproducibility in the field of LNL by reimplementing the evaluated methods and refining the benchmarking procedure.
>
> **2. Regarding concerns about the unfairness of the original work:**
>
> The original authors only made the codes for training the baseline model publicly available. Their reproduction of the evaluated methods is not public, and our work raises some concerns regarding whether the authors in fact reimplemented the methods or only used the original codebases. This concern is mainly raised by the discrepancies we observed when reproducing their benchmark experiments and the fact that their public baseline code does not produce the results described in the paper. We discuss this in Appendix A and check our results against publicly available data.
>
> Nevertheless, the original work claims to benchmark the methods in a fair manner. However, their claims are undermined by two main factors: (i) The described experiment setup (fixed training epochs and learning rate schedulers…) may adversely affect some of the methods as they optimize different loss functions. (ii) For some of the methods, they utilize different learners, affording them special treatment. The key idea of LNL is developing algorithms for training neural networks that are robust to noise. The differences in performance between two algorithms should therefore not be attributed to using a learner with higher capacity. Our improved benchmark addresses both of these issues by (i) using each method's original experiment setup  and (ii) only fixing the learner. Additionally, we also utilize noisy validation sets to replace the clean test set, which, as the reviewer noted, is not inherently problematic. However, for further method development, directly using the clean test set facilitates overfitting to the test set, and our approach (having multiple random noisy validation sets) makes this more difficult and gives a more realistic estimate of the method’s performance on unseen clean data.

---

### Review · Reviewer_W6XX · 2025-03-28

**Summary Of Contributions:**

1. The authors reproduce the baselines in the paper [1]. The detailed corresponding hyper-parameters and network structures used for reproducibility are also stated in the paper.
2. The authors conduct hypothesis testing to confirm the claim that human noisy labels in CIFAR-N differ from synthetic ones.
3. The authors reproduce the memorization behavior of the neural network on the artificial and real-world noisy data. The experiment results reveal that the model memorizes the human noisy labels faster than the synthetic ones.

**Reference**

[1] Learning with noisy labels revisited: A study using real-world human annotations. In *International Conference on Learning Representations*, 2022.

**Audience:**

Yes

**Broader Impact Concerns:**

This paper does not have ethical concerns.

**Claims And Evidence:**

Yes

**Requested Changes:**

1. Detailed experiment settings need to be described, e.g., in the sentence: "specifically the Random1 label set and its synthetic counterpart". How do you obtain the synthetic counterpart? Keep the same noise rate, then synthesize noisy data? If so, what noise type is employed?
2. The methods in the original paper (Wei et al., 2022) are out-of-date. As a paper submitted in 2025, more new methods should be included:

- Disc: Learning from noisy labels via dynamic instance-specific selection and correction
- Early Stopping Against Label Noise Without Validation Data
- Cs-isolate: Extracting hard confident examples by content and style isolation
- Late stopping: Avoiding confidently learning from mislabeled examples
- Learning the Latent Causal Structure for Modeling Label Noise
- Instance-dependent Early Stopping

3. In the sentence, the authors state that "Instead of testing every epoch, we use 5% of the noisy training data as a validation set to calculate accuracy.", which raises a concern. Specifically, some methods model the label noise using the transition matrix (Patrini et al., 2017); the accuracy of the noisy validation set can be used to select models since the clean class posterior and noise class posterior can be linked by using the transition matrix. However, many methods do not model the label noise. If the model overfits the label noise, the accuracy on the validation set will be high, but the accuracy on the test set will be low.
4. The authors did not report the noise type used in Section 4.2 when synthesizing noisy data. Whether the speed of memorizing noisy labels is related to the noise type? Will the model memorize noisy labels on pair-flipping noise faster than that on symmetry-flipping one? The author can include more results on different noise types.

**Strengths And Weaknesses:**

**Strengths**:

1. The paper reproduces the experiments on the paper [1] and verifies the claims in the original paper.
2. A fair benchmark is reported. In this benchmark, ResNet-34 is used as the backbone.

**Weaknesses**:

1. Some experiment settings are unclear in this paper, such as the noise type used to synthesize the label noise.
2. The methods in the paper [1] are out-of-date. The author should extend the benchmark (Fair LNL Benchmark) by introducing more recent methods.

**Reference**

[1] Learning with noisy labels revisited: A study using real-world human annotations. In *International Conference on Learning Representations*, 2022.

---

> ### Author Response · Authors · 2025-04-13
>
> We thank the reviewer for their time and comments, which we believe have helped improve the quality of the manuscript. We have addressed the concerns regarding missing experimental details, the use of recent state-of-the-art LNL methods, and the evaluation strategy used in our benchmark.
>
> **Clarification of Experimental Details**
>
> We appreciate the reviewer pointing out the lack of clarity around synthetic noise generation. We have added the following description to the updated version of the manuscript:
>
> > We generate the synthetic label set by sampling from an asymmetric transition matrix estimated from the CIFAR-N noisy labels. Following the approach in the original paper, we compute this matrix by counting class-wise label flips between the human-annotated (noisy) labels and the clean labels. The resulting transition matrix is then used to synthesize new noisy labels, producing a synthetic counterpart following the same transition matrix but without instance dependence.
>
> This information has also been clarified in Sections 3.3 and 4.2.
>
> **Inclusion of Recent LNL Methods**
>
> Our main goal for the paper was a reproduction of the results of the original work. Our framework is, however, extendable, therefore we welcome the reviewer’s comment. We have reviewed the methods suggested by the reviewer and wider. Out of the suggested methods, we have included the first one (DISC [2]). Out of the remaining five, two (LabelWave [3] and Instance-dependent Early Stopping [4]) are indeed related to LNL but are not applicable to our evaluation scenario (are not new LNL methods but extensions of training procedures). The other three are suitable, but two (CS-Isolate[5] and CSGN[6]) are based on DivideMix, which is (as we have shown) a very resource-intensive method, and the third one (Late Stopping [7]) is based on training multiple models. At the moment, this is (in relation to the other experiments that we have run in this revision) outside of our computing budget in the given time frame. We have additionally included the ProMix method [1], which is currently listed as top-performing on a public LNL leaderboard (Papers With Code). We hope that this inclusion of two newer methods sufficiently demonstrates the extendability of our framework.

---

> > ### Author Response · Authors · 2025-04-13
> >
> > **Requested Changes**
> > 1. We added the descriptions for how the transition matrices were obtained in the updated manuscript.
> > 2. We reimplemented two state-of-the-art LNL methods, ProMix [1] and DISC [2] and added the results to the updated manuscript.
> >
> > 3. We use a noisy validation set in our benchmark to reflect realistic LNL scenarios, where clean validation labels are often unavailable. Our intention is not to simulate an idealized setting, but rather one where robust generalization under noisy supervision is the key objective.
> > If a method performs well on a noisy validation set but generalizes poorly to clean data, this indicates poor robustness to label noise—and our benchmark penalizes such behavior by design. Conversely, robust methods should generalize well across both noisy and clean labels.
> > We also empirically show that accuracy on noisy validation data correlates well with test set performance. We attribute this to the i.i.d. nature of the training and validation splits: since both are drawn from the same noisy distribution and the model only sees the training split, overfitting to the validation set is unlikely.
> > The main caveat in our current setup would arise if a method assumes a clean validation set and uses it for more than just checkpoint selection. In such cases, we believe it is important to adapt these methods to work with noisy validation data—a scenario that reflects real-world constraints more accurately.
> > We thank the reviewer for highlighting LabelWave [3], which could potentially be integrated into future iterations of our benchmark as an alternative to traditional validation-based model selection. We have added this as a direction for future work in the revised manuscript (Appendix B).
> >
> > 4. In all of the experiments where we compare synthetic to human noise, we use the noise obtained from the same asymmetric transition matrix as the human noise. We clarified this in the manuscript, referencing the description from point 1. Based on the reviewer's suggestions, we include memorization experiments with pair-flipping and symmetric noise types. While we observe different memorization dynamics for these noise types, the human label noise still leads to faster memorization. These additional experiments are presented in Appendix C of the revised manuscript.
> >
> > [1] Xiao, Ruixuan, et al. "ProMix: combating label noise via maximizing clean sample utility." Proceedings of the Thirty-Second International Joint Conference on Artificial Intelligence. 2023.
> > [2] Li, Yifan, et al. "DISC: Learning from Noisy Labels via Dynamic Instance-Specific Selection and Correction." 2023 IEEE/CVF Conference on Computer Vision and Pattern Recognition (CVPR). IEEE Computer Society, 2023.
> > [3] Yuan, Suqin, Lei Feng, and Tongliang Liu. "Early Stopping Against Label Noise Without Validation Data." ICLR. 2024.
> > [4] Yuan, Suqin, et al. "Instance-dependent Early Stopping." arXiv preprint arXiv:2502.07547 (2025).
> > [5] Lin, Yexiong, et al. "Cs-isolate: Extracting hard confident examples by content and style isolation." Advances in Neural Information Processing Systems 36 (2023): 58556-58576.
> > [6] Lin, Yexiong, Yu Yao, and Tongliang Liu. "Learning the latent causal structure for modeling label noise." Advances in Neural Information Processing Systems 37 (2024): 120549-120577.
> > [7] Yuan, Suqin, Lei Feng, and Tongliang Liu. "Late stopping: Avoiding confidently learning from mislabeled examples." Proceedings of the IEEE/CVF international conference on computer vision. 2023.

---

### Review · Reviewer_uWYo · 2025-04-01

**Summary Of Contributions:**

The authors present a reproducibility study of the effects of learning with label noise, based on previous work. They provide a comprehensive overview of previous results along with appropriate hyperparameters. They also study the difference between synthetic noise and real (human annotation) noise on learning, using the CIFAR-N dataset.

**Audience:**

Yes

**Broader Impact Concerns:**

No ethical concernt, the paper focuses on correctly reproducing previous work.

**Claims And Evidence:**

Yes

**Requested Changes:**

It would benefit to have an additional dataset, maybe even tabular, which is not computationally expensive to confirm these insights of real vs synthetic noise are consistent. Moreover, the link between different hyperparameter and memorization could be made stronger.

**Strengths And Weaknesses:**

Strength
1. The authors are able to reproduce previous work and highlight the discrepancies in hyperparameters used, while being largely consistent.
2. The authors find that learning dynamics significantly differ for real (human) noise vs synthetic noise, human noise being learnt much faster.

Weakness
1. Since the evaluation is only on CIFAR-N, it is difficult to evaluate the differences between synthetic and real noise.
2. It seems that the different hyperparameters and the noise significanlty affects the learning dynamics. This includes memorization (overfitting) on the noisy samples. Can the authors comment on the different learning rates, weight decay and how these hyperparams affect memorization. This link is not entirely clear.

---

> ### Author Response · Authors · 2025-04-13
>
> We thank the reviewer for their thoughtful and constructive feedback. Below, we address the two main concerns raised: the limited dataset scope and the relationship between hyperparameters and memorization.
>
> **On the Use of a Single Dataset (CIFAR-N)**
>
> We appreciate the reviewer’s suggestion to include an additional dataset. We reviewed available datasets relevant to learning with noisy labels and found none that contain real-world human annotation noise paired with clean ground-truth labels, which are essential for comparing human and synthetic label noise.
> We did not manage to find any tabular datasets with real-world annotation noise. We considered ANIMAL10-N as a potential alternative; however, it unfortunately lacks clean labels. This prevents meaningful analysis of memorization dynamics or the construction of synthetic noise based on matched transition matrices. To our knowledge, CIFAR-N remains the only dataset that enables a systematic, controlled comparison of synthetic and human annotation noise.
> We agree that having additional datasets would strengthen the generalizability of our findings. As such, we are happy to include a call to the community in the revised manuscript, encouraging the development and release of new datasets with both clean and human-labeled noisy annotations.
> That said, our framework is fully extensible, and once such datasets become available, they can be easily integrated for further studies and noise comparison experiments.
>
> **On the Relationship Between Hyperparameters and Memorization**
>
> We have added additional experiments on noise memorization using different hyperparameters and synthetic noise types. Specifically, we rerun the experiments by changing synthetic labels to be generated via symmetric and pair-flipping transition matrices. For the hyperparameters, we check the impact of using a different learning rate schedule, weight decay, and learning rate values. While the memorization dynamics change when using these alternative setups, models trained on real-world noisy labels still start overfitting faster than those on synthetic ones, which further supports the claims of the original authors.
> While increasing weight decay does not noticeably impact the memorization speed, lowering the learning rate causes faster memorization for both real-world and synthetic noises (the memorization curves shift to the left). Changing the learning rate scheduler changes the shape of the memorization curves according to the learning rate schedule. The main finding, that human noise results in faster memorization than synthetic noise, stays consistent across all experiments. Therefore, we do not deem the choice of hyperparameters to be of critical importance. This is also consistent with the experiments by Xie, et al. [1], who show that SGD memorizes a high proportion of noisy labels across different hyperparameters. We hope this extended analysis strengthens the reviewer’s confidence in the generality of our findings.
>
> **Requested Changes (Summary of Updates)**
>
> We clarified in the manuscript how synthetic labels are generated using transition matrices derived from human-labeled noise (as in the original work).
>
>
> We added additional memorization experiments using:
> - Different synthetic noise types (symmetric and pair-flipping),
> - Alternative hyperparameter settings (learning rate, scheduler, and weight decay).
>
> [1] Xie, et al. "Artificial neural variability for deep learning: On overfitting, noise memorization, and catastrophic forgetting." Neural computation 33.8 (2021): 2163-2192.

---

### Decision · Action_Editor_w6YT · 2025-05-27

**Recommendation:** Reject

**Comment:**

While the paper addresses a relevant topic in noisy label learning, the empirical evaluation significantly undermines its contribution. The benchmarking relies heavily on older methods and omits many competitive recent approaches, limiting the range of audience. The authors acknowledge this limitation due to computational constraints, which is not very convincing. To the best of our knowledge, most learning-with-noisy-label methods can be run on a single 3090 GPU.

**Audience:**

It is unlikely that the findings of this paper will generate substantial interest within the TMLR audience. The problem setting and general approach have already been well studied in prior literature, and the paper does not offer sufficiently novel insights or techniques to re-engage the community. In addition, the empirical evaluations do not clearly demonstrate current state-of-the-art, limiting the paper’s potential impact.

**Claims And Evidence:**

The claims made in the submission are not convincingly supported by up-to-date or comprehensive evidence. While the authors conduct an empirical evaluation, the choice of benchmark methods is a major weakness. The majority of baselines used in the comparison are outdated, with only two recent methods from 2023 included. Only two small-scale datasets CIFAR10N CIFAR100N are used. Given the fast pace of research in this area, such a selection significantly limits the strength and relevance of the empirical claims.

**Resubmission Of Major Revision:**

The authors may consider submitting a major revision at a later time.

---

> ### Author Response · Authors · 2025-06-02
> **Clarification Regarding Submission Type and Evaluation Criteria**
>
> Thank you for your timely decision. Given the nature of our submission and the review process to date, we were somewhat surprised by the rejection, as it does not appear to align with the core TMLR acceptance criteria of claims and audience, both of which received positive assessments from all three reviewers.
>
> The main points cited in the decision (the lack of novelty, reliance on older baselines, and use of small-scale datasets) seem more applicable to original research papers than to reproducibility studies. We would like to respectfully clarify that our submission was made under the latter category, as outlined in the TMLR guidelines. The primary objective of our work is to rigorously replicate and analyze the behavior of the original paper, with a specific focus on evaluating performance under real-world, human annotation noise using the CIFAR-N datasets.
>
> As part of this effort, extending beyond the original paper, we re-implemented several widely used LNL methods in a unified framework, reproduced the key results, and conducted additional experiments to deepen the analysis. Our intention is to release this framework to the community to support further reproducibility and extension.
>
> Two reviewers suggested possible extensions to the study, which we did our best to address during the revision window. We have included two additional LNL methods, one of which is currently SOTA in the field according to public leaderboards. Given additional time, we would have been willing to implement more recent methods, provided they fit within the given evaluation scenario. However, we found no suitable alternative datasets that offer the same kind of structured, human annotation noise as CIFAR-N. We believe this scarcity reinforces the relevance of CIFAR-N and justifies its use as a testbed for reproducibility-focused studies.
>
> Since the reviewers did not raise major concerns regarding the correctness, clarity, or fidelity of our work, we would appreciate it if you could confirm whether the submission was evaluated under the appropriate category and according to the intended criteria for reproducibility studies. If there has been a misunderstanding, we would welcome the opportunity to clarify or address it.

---

> > ### Comment · Action_Editor_w6YT · 2025-07-22
> > **Followup Comments**
> >
> > We all believe that comparing the recent or SOTA methods for learning with noisy labels under a fair setting is very important. Here, I would like to list some suggestions to make this paper stronger.
> >
> > The major concern for this paper is that the claims are not interesting to the audience because of the limited experiements.
> >
> > Specifically, the majority of baselines used are outdated, with only two recent methods from 2023 included.
> > Only two small-scale toy datasets (CIFAR-10N and CIFAR-100N) are used, which is not sufficient to illustrate the method’s effectiveness in real-world, large-scale settings.
> > As a reproducibility paper, the number of experiments is less than the average ICML/ICLR/CVPR conference paper (about 1/4 to 1/3).
> > The evaluated methods have already been well compared in the literature under the same settings and on these datasets, so there are few new insights.
> >
> > To make the paper acceptable, it should add additional insights by fairly comparing recent methods in the current setting. The large-scale benchmark datasets for label-noise learning and all related papers are listed as follows from ICML, ICLR, and NeurIPS under the traditional label-noise setting since 2023.
> >
> > Specifically, 1) The authors need to compare at least half of these methods; 2) The evaluation should include at least two real-world large-scale benchmark datasets.
> >
> > ----------Attached datasets and papers----------
> >
> > **Datasets:**
> > Animal-10N
> > Food-101N
> > WebVision
> > Clothing1M
> >
> > **List of papers:**
> >
> > **ICLR 2025**
> >
> > * Instance-dependent Early Stopping
> >
> > **ICLR 2024**
> >
> > * Early Stopping Against Label Noise Without Validation Data
> > * Dirichlet-based Per-Sample Weighting by Transition Matrix for Noisy Label Learning
> > * Robust Classification via Regression for Learning with Noisy Labels
> >
> > **ICML 2024**
> >
> > * Pi-DUAL: Using Privileged Information to Distinguish Clean from Noisy Labels
> >
> > **NeurIPS 2024**
> >
> > * ε-Softmax: Approximating One-Hot Vectors for Mitigating Label Noise
> > * Learning the Latent Causal Structure for Modeling Label Noise
> > * Label Noise: Ignorance Is Bliss
> > * Learning from Noisy Labels via Conditional Distributionally Robust Optimization
> > * Noisy Label Learning with Instance-Dependent Outliers: Identifiability via Crowd Wisdom
> >
> > **NeurIPS 2023**
> >
> > * Active Negative Loss Functions for Learning with Noisy Labels
> > * CSOT: Curriculum and Structure-Aware Optimal Transport for Learning with Noisy Labels
> > * CS-Isolate: Extracting Hard Confident Examples by Content and Style Isolation
> > * Label-Retrieval-Augmented Diffusion Models for Learning from Noisy Labels

---

> ### Comment · Action_Editor_w6YT · 2025-06-02
>
> Dear Authors,
>
> We sincerely apologize for the confusion. The major concerns are not about novelty but rather the number of baselines and datasets included for comparison. Note that Reviewer W6XX assigned yes by mistake but cannot update the option in the system
>
> As a benchmark paper, it should cover more methods and datasets than an average conference paper. However, the depth and breadth of the experiments are less than those in recent papers from ICML, NeurIPS, and ICLR.  Specifically, the majority of baselines used in the comparison are outdated, with only two recent methods from 2023 included, and only two small‐scale datasets (CIFAR10N and CIFAR100N) used.
>
> We believe that this can greatly limit the audience attraction.
>
> This is not a discouragement, we are trying to improve this work to make it more impactful. We believe that a benchmark paper evaluated under fair settings is highly valuable. Therefore, please make use this chance and consider submitting a major revision at a later date.
>
> Best,
> AE